# Fucoidan-Manganese Dioxide Nanoparticles Potentiate Radiation Therapy by Co-Targeting Tumor Hypoxia and Angiogenesis

**DOI:** 10.3390/md16120510

**Published:** 2018-12-15

**Authors:** Sung-Won Shin, Wooju Jung, Changhoon Choi, Shin-Yeong Kim, Arang Son, Hakyoung Kim, Nohyun Lee, Hee Chul Park

**Affiliations:** 1Department of Radiation Oncology, Samsung Medical Center, Seoul 06351, Korea; camuserik@gmail.com (S.-W.S.); chchoi93@gmail.com (C.C.); syeong.kim@sbri.co.kr (S.-Y.K.); arang.son@sbri.co.kr (A.S.); khk614@gmail.com (H.K.); 2School of Medicine, Sungkyunkwan University, Seoul 06351, Korea; 3School of Advanced Materials Engineering, Kookmin University, Seoul 02707, Korea; jwj139@daum.net

**Keywords:** tumor hypoxia, radiation therapy, nanoparticles, fucoidan

## Abstract

Tumor hypoxia is a major mechanism of resistance to radiation therapy (RT), which is associated with poor prognosis in affected cancer patients. Various approaches to treat hypoxic and radioresistant cancers, including pancreatic cancer, have shown limited success. Fucoidan, a polysaccharide from brown seaweed, has antitumor and antiangiogenesis activities. Here, we discuss the development of fucoidan-coated manganese dioxide nanoparticles (Fuco-MnO_2_-NPs) and testing of the therapeutic potential with RT using pancreatic cancer models. In vitro data showed that Fuco-MnO_2_-NPs generated oxygen efficiently in the presence of H_2_O_2_ and substantially suppressed HIF-1 expression under a hypoxic condition in human pancreatic cancer cells. Fuco-MnO_2_-NPs reversed hypoxia-induced radioresistance by decreasing clonogenic survival and increasing DNA damage and apoptotic cell death in response to RT. In a BxPC3 xenograft mouse model, the combination treatment with Fuco-MnO_2_-NPs and RT resulted in a greater tumor growth delay than RT alone. Fucoidan-coated NPs, but not naked ones, further suppressed tumor angiogenesis, as judged by immunohistochemistry data with diminished expression of phosphorylated vascular endothelial growth factor receptor 2 (VEGFR2) and CD31. These data suggest that Fuco-MnO_2_-NPs may potentiate the effects of RT via dual targeting of tumor hypoxia and angiogenesis, and they are of great clinical potential in the treatment of hypoxic, radioresistant pancreatic cancer.

## 1. Introduction

Pancreatic cancer is the fourth most common cause of cancer-related deaths and was estimated to be responsible for 43,090 deaths in the United States in 2017 [1,2]. The most common type, pancreatic adenocarcinoma, typically has a very poor prognosis. Following diagnosis, 25% of people survive for one year and 5% live for five years. For cancers diagnosed early, the five-year survival rate rises to about 20% [3]. Surgical removal of the tumor is the only cure for this cancer. Whether or not surgical resection can be offered depends on the cancer spread and the proximity of the major blood vessels to the pancreas. Unfortunately, surgical resection is only possible in approximately 20% of new cases. Chemotherapy or chemoradiotherapy is likely to be offered to most patients with advanced pancreatic cancer. However, the changes made in the last few years have only increased survival times by a few months [4], and the high rate of treatment failure remains a challenge [5].

Radiation therapy (RT) may constitute a part of the treatment to shrink tumors to a resectable state, but its use on unresectable tumors remains controversial due to conflicting results from previous studies [6]. Regarding radiation physics, recent advances in radiation delivery techniques, such as intensity-modulated radiotherapy and image-guided radiotherapy, have contributed to the preservation of normal tissue, which enables a decrease in normal tissue toxicity while increasing local control rate through safe dose-escalation [7]. In terms of radiation biology, oxygen is an important factor in determining the radiosensitivity of tumor cells. The presence or absence of oxygen dramatically influences the biologic effects of X-rays. Oxygen can enhance and fix radiation damage such as DNA double-strand breaks (DSBs), causing cancer cells to undergo apoptosis [8,9]. In the absence of oxygen, damage produced by the indirect action may be repaired. In addition, the importance of rapid and complete reoxygenation is well-known because hypoxic cells have little influence on the outcome of a fractionated radiation schedule. In addition, oxygen can also regulate radiosensitivity by altering intracellular gene expression. Tumor hypoxia leads to the activation of a transcription factor, called hypoxia-inducible-factor-1 (HIF-1). HIF-1α is involved in resistance to radiotherapy, chemotherapy and metastasis through the regulation of cell death, DNA damage repair, cell proliferation, and angiogenesis [10,11]. Therefore, in hypoxic tumors, the therapeutic efficacy of radiotherapy is decreased.

To date, various strategies such as hyperbaric oxygen therapy and drugs that preferentially kill or sensitize hypoxic cells to radiation [12] have been proposed to overcome hypoxia-induced radiation resistance. Unfortunately, side effects, such as increased reactive oxygen species (ROS) generation and off-target toxicity, limit clinical applications. Multifunctional nanoplatforms are attracting increasing attention in this regard. Albumin-coated manganese dioxide nanoparticles (A-MnO_2_-NPs) were developed to overcome tumor hypoxia; they can generate oxygen through decomposition of endogenous hydrogen peroxide (H_2_O_2_) and increase pH. Modulation of the tumor microenvironment by A-MnO_2_-NPs can enhance the RT response [13]. Further development of MnO_2_-NPs has been performed for tuning oxygen generation rates and optimizing colloidal stability under physiological conditions, which has resulted in the marked potentiation of RT in murine tumor models [14]. MnO_2_-NPs conjugated with acriflavine, a potent HIF-1 inhibitor, were also recently developed and shown to enhance RT response and abscopal effects, suggesting the great potential of MnO_2_-based nanoplatforms as radiation sensitizers for treating metastatic as well as primary tumors [15].

Seaweeds are rich in biologically active polysaccharides that exhibit a broad spectrum of biological activities. Fucoidan is a fucose-rich, sulfated cell-wall polysaccharide isolated from brown seaweed. Fucoidan has diverse activities, including anticancer and anticoagulant activities [16]. Recent studies have suggested that fucoidan can act as an anticancer agent through the modulation of the immune system [17,18]. Fucoidan induces the maturation of dendritic cells and, in association with other cytokines, shapes the immune responses that are mediated by T-cells. Fucoidan also has an inhibitory role on cancer metastasis and angiogenesis via the downregulation of matrix-metalloproteinase 2 and the vascular endothelial growth factor (VEGF) [19]. In this study, we newly designed fucoidan-manganese dioxide nanoparticles (Fuco-MnO_2_-NPs) with dual functionality by conjugation with bioactive fucoidan and oxygen-generating MnO_2_-NPs. Subsequently, we investigated the preclinical potential of the use of a Fuco-MnO_2_-NP formulation in a pancreatic cancer model. The aim was to meet the urgent need for novel hypoxic tumor radiosensitizers.

## 2. Results

### 2.1. Synthesis of Fuco-MnO_2_-NPs

Fuco-MnO_2_-NPs were synthesized according to a previous report, with slight modifications [13]. Manganese permanganate (KMnO_4_) was reduced in the presence of poly (allylamine hydrochloride) (PAH) at room temperature, resulting in stable MnO_2_-NPs. Figure 1a shows a transmission electron microscopy (TEM) image of MnO_2_-NPs. Owing to the small core size and low crystallinity, the morphology of MnO_2_-NPs was not clearly revealed. However, ultraviolet-visible (UV-Vis) spectra showed the disappearance of KMnO_4_ peaks around 315 nm and 530 nm and the appearance of a broad peak around 300 nm, representing the surface plasmon band of colloidal MnO_2_ (Figure 1b). The hydrodynamic size and zeta-potential measurements of PAH-coated NPs revealed an over 17 nm in overall size and +55.1 mV in surface charge (Figure 1c,d). Since a strong positive surface charge may induce cytotoxicity, the surfaces of the nanoparticles were modified with negative fucoidan using electrostatic interaction. After fucoidan absorption, the overall size and surface charge of Fuco-MnO_2_-NPs were changed to −48.61 nm and −24.6 mV, respectively (Figure 1c,d). No severe aggregation was observed after fucoidan coating.

The reactivity of the Fuco-MnO_2_-NPs was evaluated using time-dependent H_2_O_2_ assays after the addition of NPs. Within 60 min, 2.5 mM of H_2_O_2_ was almost completely decomposed by 250 µM of Fuco-MnO_2_-NPs (Figure 1e). Then, the generation of O_2_ at a low concentration of H_2_O_2_ (250 µM) was evaluated under a hypoxic condition. Owing to the high activity of the Fuco-MnO_2_-NPs, a significant amount of oxygen was detected (Figure 1f). These results indicate that fucoidan coating does not affect the reactivity of MnO_2_-NPs toward H_2_O_2_.

### 2.2. Fuco-MnO_2_-NPs Radiosensitized Pancreatic Cancer Cells in Vitro

To determine the optimal nanoparticle dosage, the cytotoxicity of Fuco-MnO_2_-NPs was assessed in human pancreatic cancer cell lines (AsPC-1 and BxPC-3) using a cell counting kit 8 (CCK-8; Dojindo, Mashiki-machi, Japan) assay (Figure 2a). Both PAH-MnO_2_-NPs and Fuco-MnO_2_-NPs inhibited the cell proliferation of AsPC-1 and BxPC-3 in a dose-dependent manner. Notably, Fuco-MnO_2_-NPs were slightly less toxic than PAH-MnO_2_-NPs at 5 μg/mL and 10 μg/mL, although the difference was not significant (Figure 2a).

Next, to determine the radiosensitizing effect of Fuco-MnO_2_-NPs on hypoxic tumor cells, we devised a system in which hypoxic tumor cells are irradiated using a hypoxia chamber (Figure 2b). The establishment of a hypoxic tumor cell culture condition was confirmed by measuring the increased expression of HIF-1α using Western blot analysis (Figure 2c), and increased luminescence using a hypoxia-responsive luciferase activity assay (Figure 2e). Treatment with Fuco-MnO_2_-NPs and PAH-MnO_2_-NPs, but not fucoidan alone, clearly suppressed hypoxia-induced HIF-1α expression (Figure 2c,d) or hypoxia-responsive luciferase activity (Figure 2e) in pancreatic cancer cells, indicating that the Fuco-MnO_2_-NPs alleviated the hypoxic condition via the generation of oxygen.

The sensitization of the pancreatic cancer cells to radiation by Fuco-MnO_2_-NPs was assessed by a clonogenic assay using the hypoxia chamber system described above (Figure 2f). The survival of AsPC-1 and BxPC-3 cells was slightly higher after treatment with ionizing radiation (IR) of 4 Gy X-rays under the hypoxic condition than under a normoxic condition, which suggests a hypoxia-mediated radiation resistance phenotype. Fuco-MnO_2_-NPs and PAH-MnO_2_-NPs decreased the clonogenic survival of AsPC-1 and BxPC-3 cells under hypoxic condition in response to IR (Figure 2f), indicating that these NPs resensitized radioresistant pancreatic cancer cells by relieving hypoxia.

### 2.3. Fuco-MnO_2_-NPs Enhanced Radiation-Induced Apoptosis in a Hypoxic Condition

We examined the effects of Fuco-MnO_2_-NPs on IR-mediated cell-cycle redistribution using flow cytometry (Figure 3a). A DNA content analysis with propidium iodide (PI) staining in BxPC-3 cells revealed that cell populations at sub-G1, but not other cell cycle phases, increased at 72 post-irradiation; this effect was slightly weaker in the hypoxic condition than in the normoxic condition (Figure 3a). Fuco-MnO_2_-NPs and PAH-MnO_2_-NPs, but not fucoidan alone, slightly increased IR-induced sub-G1 populations, even though the difference was not statistically significant. Given that the induction of the sub-G1 peak may represent DNA fragmentation during apoptosis, we determined the impact of Fuco-MnO_2_-NPs on IR-induced apoptosis using flow cytometry with Annexin V and PI double staining (Figure 3b). 6 Gy of IR dramatically increased the apoptotic cell population under the normoxic condition, and this effect was suppressed by hypoxic culture (28.7% vs. 15.0%, *p* < 0.005). The combination treatment with Fuco-MnO_2_-NPs and IR significantly increased the percentage of apoptotic cells compared with IR alone (15.0% vs. 25.2%, *p* < 0.05) under the hypoxic condition, but not under the normoxic condition.

### 2.4. Fuco-MnO_2_-NPs Inhibited DSB Repair and HIF-1α Expression under Hypoxia

To elucidate the underlying mechanisms of radiosensitization by Fuco-MnO_2_-NPs, we determined the repair kinetics of IR-induced DNA damage. The DNA repair was evaluated by measuring the foci of phosphorylated histone H2AX (γ-H2AX), which is a sensitive indicator of DSB. γ-H2AX foci were clearly seen within cell nuclei at 2 h post-irradiation, and some of the foci had disappeared at 24 h post-irradiation owing to the repair process (Figure 4a, upper panel). The number of γ-H2AX foci at 24 h post-irradiation was less in the hypoxia than under the normoxia condition, indicating an oxygen effect on DNA damage repair (Figure 4a, lower panel). PAH-MnO_2_-NPs or Fuco-MnO_2_-NPs increased the numbers of γ-H2AX foci at 24 h post-irradiation compared to the saline control or fucoidan (Figure 4a, right panel). Thus, our data suggest that Fuco-MnO_2_-NPs may attenuate DNA damage repair through increased oxygen effect, resulting in radiosensitization in BxPC-3 cells.

HIF-1α is a key mediator of hypoxia signaling [20,21] and is involved in hypoxia-induced radioresistance. We examined the effect of Fuco-MnO_2_-NPs on HIF-1α expression by flow cytometry and immunofluorescence imaging. The BxPC-3 cells were preincubated with PAH-MnO_2_-NPs or Fuco-MnO_2_-NPs in a normoxic or hypoxic condition for 4 h, followed by 4 Gy of IR. The expression of HIF-1α was determined at 24 h post-irradiation. HIF-1α expression was induced by IR alone, although to a much smaller degree than by hypoxia. Hypoxia markedly increased the percentage of HIF-1α-positive cells (Figure 4b) and the nuclear expression of HIF-1α (Figure 4c) in comparison with normoxia. Both PAH-MnO_2_-NPs and Fuco-MnO_2_-NPs significantly suppressed hypoxia-induced HIF-1α expression in BxPC-3 cells (Figure 4b,c), suggesting that MnO_2_-NPs may reduce radioresistance by suppressing hypoxia-induced HIF-1α signaling. HIF1-1α expression also decreased, but to a much lesser extent than did MnO_2_-NPs.

### 2.5. Fuco-MnO_2_-NPs Radiosensitized Pancreatic Cancer Cells In Vivo

The in vivo radiosensitization effect of Fuco-MnO_2_-NPs in pancreatic cancer was confirmed using BxPC-3 xenograft tumor models. The effects of IR and its combination with Fuco-MnO_2_-NPs on tumor growth were evaluated using BALB/c nude mice bearing BxPC3 xenograft tumors. Intratumoral injection with Fuco-MnO_2_-NPs did not change BxPC-3 tumor growth, whereas a single 8 Gy dose of IR significantly inhibited tumor growth (Figure 5a,b). The combination of Fuco-MnO_2_-NPs and IR further delayed tumor growth compared with IR alone (Figure 5a,b). On day 30, the average tumor volume in the co-treated group was reduced by 46.4% in comparison with the control group and further reduced by 22.5% compared with the IR-alone group (Figure 5b; *p* < 0.05). Fucoidan alone did not change tumor growth, but Fuco-MnO_2_-NPs did, although not in a statistically significant manner (data not shown). Unusual symptoms, such as sickness or body weight loss in mice, were not seen after injection of Fuco-MnO_2_-NPs, IR at 8 Gy or their combination (Figure 5c).

### 2.6. Fuco-MnO_2_-NPs Induced Apoptosis and Inhibited Angiogenesis in Irradiated BxPC-3 Xenograft Tumors

Based on our in vitro results of the enhancement of IR-induced apoptosis by Fuco-MnO_2_-NPs, we determined the effects of Fuco-MnO_2_-NPs on apoptosis in BxPC-3 xenograft tumor tissues. For this, we performed terminal deoxynucleotidyl transferase dUTP nick end-labeling (TUNEL) assay. TUNEL-positive cells were increased in the IR-treated groups and were further enhanced by either Fuco-MnO_2_-NP or PAH-MnO_2_-NP co-treatment (Figure 6a,b). This is consistent with in vitro annexin V staining data (Figure 3b). Fucoidan alone did not affect TUNEL staining results. Immunohistochemistry (IHC) analysis showed that HIF-1α expression was greatly decreased in both Fuco-MnO_2_-NP and PAH-MnO_2_-NP cotreated groups (Figure 6a,c), suggesting that MnO_2_-NPs may alleviate tumor hypoxia in vivo as well as in vitro.

As VEGF/VEGF receptor (VEGFR) signalling is a key factor of tumor angiogenesis and radiation resistance, we investigated the effects of Fuco-MnO_2_-NPs on VEGF-induced phosphorylation of VEGFR2 in BxPC-3 tumor tissue using IHC staining. Phospho-VEGFR2 expression was reduced by MnO_2_-NPs or fucoidan (Figure 6a,d). The Fuco-MnO_2_-NPs-treated group displayed 27.0% greater reduction of phospho-VEGFR2 expression compared with the MnO_2_-NPs-treated group. We further examined tumor vasculature with cluster of differentiation 31 (CD31) IHC staining. The expression of CD31, a vascular endothelial marker, showed minimal change following treatment with MnO_2_-NPs or fucoidan, but more marked reduction was seen in Fuco-MnO_2_-NPs-treated tumors (Figure 6a,e). As fucoidan was previously shown to inhibit angiogenesis by downregulating HIF-1α/VEGF signalling under hypoxia [22], it is reasonable to suggest that Fuco-MnO_2_-NPs were more effective at inhibiting angiogenesis than PAH-MnO_2_-NPs.

## 3. Discussion

RT with concurrent chemotherapy is the mainstay treatment for advanced pancreatic cancer. However, high recurrence rates and resistance to previous treatment remain challenges. Several mechanisms, including tumor hypoxia, DNA damage repair, inflammation, and angiogenesis, are associated with resistance to treatment. Among these, the biological impact of oxygen on the outcome of a fractionated radiotherapy session has been widely studied. Numerous efforts to overcome hypoxia-mediated resistance to RT have only had limited success. The clinical use of hypoxia-activated prodrugs, such as evofosfamide (TH-302), has been considered as a monotherapy or in combination with various chemotherapeutics, including gemcitabine, in pancreatic cancer patients [23]. The combination effect of evofosfamide with radiation was recently studied using a pancreatic tumor orthotopic mouse model [24].

The enhancement of the therapeutic efficacy of radiotherapy in hypoxic tumors could be achieved by directly supplying adequate oxygen within tumors. Hyperbaric oxygen therapy is a well-established treatment to overcome hypoxia, where pure O_2_ is supplied in a pressurized room. However, this may cause overproduction of ROS in normal tissue, resulting in serious side effects such as barotrauma and hyperoxic seizures [25]. In addition, the O_2_ molecule can be directly delivered using its high solubility for perfluorocarbon [26]. Unfortunately, the size of perfluorocarbon-based emulsion is too large, which limits penetration into a hypoxic region. One of the strategies to deliver oxygen into hypoxic tumor tissues is to use oxygen-generating nanomaterials, such as CaO_2_-NPs [27], C_3_N_4_-NPs [28], and catalase-containing emulsion [29]. Recent studies have reported that MnO_2_-NPs can convert H_2_O_2_, which is abundantly produced in tumor tissues, into oxygen, thus modulating tumor hypoxia and enhancing radiation efficacy in mouse models [13,14,15]. In these studies, MnO_2_-NPs are coated with biologically inactive albumin. To produce a hypoxic radiosensitizer in pancreatic cancer, we designed a new nanoplatform in which oxygen-generating MnO_2_-NPs were coated with bioactive marine products known as fucoidans. Fucoidans are highly sulfated cell-wall polysaccharides isolated from a species of brown seaweed. Fucoidans have diverse biological activities, including anti-inflammatory, antioxidant, and antitumor activity [16,17,18]. In nanomedicine, fucoidan coating is used to enhance anticancer effects and also to reduce the toxicity of metallic nanoparticles [30].

In the current study, we focused on the effects of Fuco-MnO_2_-NPs on RT efficacy through the modulation of a hypoxic tumor microenvironment. We first investigated the in vitro radiosensitizing activity of Fuco-MnO_2_-NPs against hypoxic pancreatic cancer cells. Fuco-MnO_2_-NPs inhibited the clonogenic survival of pancreatic cell lines, including AsPC-1 and BxPC-3, after RT under a hypoxic condition. Specifically, they enhanced radiation-induced apoptosis and attenuated DNA damage repair in BxPC-3 cells, which were suppressed by hypoxia compared with normoxia. We also obtained similar results from MIA-PaCa-2 cells (data not shown). Consistent with the in vitro data, in vivo Fuco-MnO_2_-NPs, in combination with radiation, significantly inhibited the growth of BxPC-3 xenograft tumors. TUNEL analysis confirmed that the combination of Fuco-MnO_2_-NPs and radiation greatly increased apoptosis in tumor tissues. The expression of HIF-1α, reflecting a hypoxic microenvironment, correlates with the resistance of cells to radiation, so HIF-1α is an excellent potential target for improving the efficacy of RT [31]. Fuco-MnO_2_-NPs additionally successfully reduced the HIF-1α expression in pancreatic cancer cells cultured in a hypoxic condition, indicating efficient oxygen generation. Intratumoral injection of Fuco-MnO_2_-NPs also alleviated tumor hypoxia in vivo, as supported by decreased HIF-1α expression in xenografted tumor tissues. These data suggest that Fuco-MnO_2_-NPs enhance RT efficacy via the accumulation of unrepairable DNA and induction of apoptosis, which are due to oxygenation of hypoxic tumors, as reflected by reduced HIF-1α expression.

Tumor angiogenesis helps to deliver oxygen, nutrients, and growth factors to cancer cells, leading to rapid growth and metastasis [32]. HIF-1α is a potent stimulator of neovascularization and excretion of VEGF at a transcription level, and radiation induces the upregulation of VEGF expression and angiogenesis [33,34]. Targeting the VEGF pathway inhibits new vessel development, causing tumor regression. Several studies have reported the potential antiangiogenic activity of fucoidans [22,35,36]. Fucoidan inhibits the binding of VEGF to its cell membrane receptor. The antiangiogenic properties of fucoidan were observed previously in 4T1 mouse breast cancer cells *both* in vitro *and* in vivo and were related to a significant dose-dependent decrease in VEGF expression in cells treated with fucoidan [37]. However, while fucoidan did inhibit hepatoma tumor growth, it did not suppress angiogenesis or VEGF expression. This may be because the antiangiogenic effects of fucoidans depend on their diverse molecular weights and structures [38]. Our IHC data indicate that Fuco-MnO_2_-NPs inhibited radiation-induced angiogenesis signaling, as judged by the presence of decreased expression of phospho-VEGFR2 and CD31 in BxPC-3 xenograft tumor. The antiangiogenic activity of Fuco-MnO_2_-NPs may be a combination effect of MnO_2_-mediated HIF-1α downregulation and the antiangiogenic effects of the fucoidan moiety.

Pancreatic cancer still remains a major health issue with a poor prognosis, despite the recent advance in therapeutics, and hypoxia is a major therapeutic resistance mechanism in pancreatic cancer [39]. Thus, new therapeutic strategies for overcoming hypoxia are urgently needed. The application of nanomedicine for cancer therapy has received growing attention because of many advantageous aspects, including the improvement of drug delivery to tumors. In this study, we demonstrate that Fuco-MnO_2_-NPs potentiated pancreatic tumors to radiation by alleviating hypoxia in vitro and in vivo. Our findings suggest the potential of this new nanoplatform that can dual-target hypoxia and angiogenesis in treating hypoxic tumors, including pancreatic cancer, to particularly improve RT efficacy.

## 4. Materials and Methods 

### 4.1. Materials

KMnO_4_ (ACS reagent, ≥99.0%), poly (allylamine hydrochloride) (~17,500 kDa) and fucoidan from *Fucus vesiculosus* (≥95%) were purchased from Sigma-Aldrich (St. Louis, MO, USA). The Annexin V-FITC apoptosis detection kit was purchased from BD Biosciences (San Jose, CA, USA). Anti-rabbit and anti-mouse Alexa-Fluor-488-conjugated secondary antibodies were obtained from Molecular Probes (Carlsbad, CA, USA). Anti-HIF-1α were obtained from Novus Biochemicals (Littleton, CO, USA). Antiphospho-γ-H2AX antibodies were obtained from Cell Signaling Technology (Danvers, MA, USA) and Millipore (Burlington, MA, USA). Anti-CD31 and phospho-VEGF receptor 2 antibodies ware purchased from Abcam (Cambridge, UK). The anti-β-actin, anti-rabbit immunoglobulin G, and anti-mouse immunoglobulin G secondary antibodies; PI; DAPI; and fucoidan were from Sigma-Aldrich. Unless stated otherwise, all other chemicals were purchased from Sigma-Aldrich. All acronyms used in this manuscript are listed in Table A1.

### 4.2. Synthesis of Nanoparticle

MnO_2_-NPs were prepared according to the protocols of a previous report, with a slight modification [13]. Briefly, 18 mL of KMnO_4_ solution (3.5 mg/mL) was mixed with 2 mL of PAH solution (37.4 mg/mL). The mixture was vigorously stirred until the color of the solution changed to brown. MnO_2_-NPs coated with PAH were purified using a centrifugal filter (Amicon Ultra-15, MWCO 100 kDa; Millipore). The separated nanoparticles were dispersed in 15 mL of distilled water. The concentration of Mn^2+^ in the solution was determined using an inductively coupled plasma atomic emission spectrometer (ICPS-8100; Shimadzu, Kyoto, Japan). To prepare the Fuco-MnO_2_-NPs, fucoidan was added to the MnO_2_-NP solution to produce a mass ratio of Mn^2+^ to fucoidan of 1:2, and the solution was stirred for 3 h. The morphology of the nanoparticles was observed by TEM using an EM-2010 microscope (JEOL, Tokyo, Japan) at 200 kV. The absorption spectra of KMnO_4_ and MnO_2_-NP were obtained using a UV-Vis spectrophotometer (UV-2600; Shimidazu). The hydrodynamic size and zeta-potential of the nanoparticles were measured by dynamic light scattering (Zetasizer ZS90; Malvern, Worcestershire, UK).

### 4.3. Measurement of H_2_O_2_ Decomposition and O_2_ Generation

To demonstrate H_2_O_2_ decomposition, 2.5 mM of H_2_O_2_ and 250 µM of Fuco-MnO_2_-NPs were mixed in phosphate-buffered saline (PBS). Subsequently, 50 µL of the mixture was added to 100 µL of Ti(SO_4_)_2_ solution (1.33 mL of 24% Ti(SO_4_)_2_ and 8.33 mL of H_2_SO_4_ in 50 mL of deionized water) every 30 min. The concentration of H_2_O_2_ was estimated by measuring the absorbance at 405 nm. To investigate the generation of oxygen, 250 µM of H_2_O_2_ was incubated with 250 µM of Fuco-MnO_2_-NPs. Prior to the addition of Fuco-MnO_2_-NPs, dissolved oxygen was removed by bubbling with argon. The concentration of oxygen was measured with a dissolved oxygen meter (HI9136; Hanna Instruments, Woonsocket, RI, USA).

### 4.4. Cell Culture, Generation of Hypoxic Environment, and Irradiation

AsPC-3 and BxPC-3 human pancreatic cancer cell lines were obtained from the Korean Cell Line Bank (Seoul National University, Seoul, Korea) and cultured in Roswell Park Memorial Institute 1640 medium or Dulbecco’s modified Eagle’s medium, supplemented with 10% fetal bovine serum (FBS), 100 U/mL of penicillin, 100 μg/mL of streptomycin, and 25 mM of HEPES (Gibco, Carlsbad, CA, USA). Cultures were maintained in a humidified atmosphere of 95% air and 5% CO_2_ at 37 °C.

For hypoxic cell culture, cells were placed in a Hypoxia Incubator Chamber (#27310; STEMCELL Technologies, Vancouver, BC, Canada) that was flushed with a gas mixture of 1% O_2_, 5% CO_2_, and 94% N_2_ (Danil Syschem, Seoul, Korea). After gas flushing, the chamber was kept in an incubator at 37 °C in an atmosphere of 5% CO_2_ prior to irradiation. Cells placed in the hypoxia chamber without gas flushing were used for the normoxic control.

For IR treatment, cell monolayers were irradiated with various doses of 6 MV photons at a dose rate of 3.96 Gy/min using a Varian Clinac 6EX accelerator (Varian Medical Systems, Palo Alto, CA, USA).

### 4.5. Cell Viability Assay

Cell viability was measured using a CCK-8 (Dojindo, Mashiki-machi, Japan). Briefly, cells were seeded in 96-well plates at a density of 3 × 10^3^ cells/well for 12 h. After incubation with different doses of PAH-MnO_2_-NPs and Fuco-MnO_2_-NPs for 72 h, 10 μL of CCK-8 solution was added to each well and incubated for another 2 h. The absorbance was measured at 450 nm using a SpectraMAX i3 microplate reader (Molecular Devices, San Jose, CA, USA).

### 4.6. Clonogenic Survival Assay

Radiosensitivity was determined by clonogenic survival assay, as previously described [40,41,42]. Cells were seeded in six-well plates at specific cell numbers (200–1000). The cells were pretreated with PAH-MnO_2_-NPs (1 μg/mL), fucoidan (0.5 μg/mL), and Fuco-MnO_2_-NPs (1 μg/mL) for 3 h, followed by exposure to increasing doses of IR (0, 2, 4, 6, and 8 Gy). After 7 to 14 days, cells were fixed in 98% ethanol and stained with 0.5% crystal violet (Sigma-Aldrich, St. Louis, MO, USA), and the colonies were counted using an inverted microscope (Zeiss Primovert; Carl Zeiss Co., Ltd., Jena, Germany). Colonies that contained > 50 cells were considered viable.

### 4.7. Western Blot Analysis

Cells were harvested, washed with PBS, and treated with a lysis buffer containing 20 mM Tris (pH 8.0), 137 mM of NaCl, 10% glycerol, 1% Nonidet P-40, 10 mM EDTA, 100 mM NaF, 1 mM phenylmethylsulfonyl fluoride, and 10 mg/mL leupeptin. The lysates were centrifuged at 13,000 rpm for 15 min, and the concentration of protein in each lysate was determined using Bio-Rad protein assay reagent (Bio-Rad Laboratories, Hercules, CA, USA), in accordance with the manufacturer’s recommendations. Then, 8%, 10% or 12% sodium dodecyl sulphate–polyacrylamide gel electrophoresis was used to separate 20 μg protein samples. Following electrophoresis, proteins were transferred to nitrocellulose membranes (Bio-Rad Laboratories), blocked overnight in 5% skim milk in PBS at 4 °C, and subsequently probed with a primary antibody. The blots were also probed with a monoclonal anti-β-actin antibody (Sigma-Aldrich, St. Louis, MO, USA) to be quantified as a relative loading control. The detection of specific proteins was carried out with enhanced chemiluminescence detection reagents (GE Healthcare, Marlborough, MA, USA) following the manufacturer’s instructions.

### 4.8. Cell Cycle Analysis

In this step, 2 × 10^5^ cells were plated in six-well plates and allowed to attach overnight. Then, the cells were pretreated with PAH-MnO_2_-NPs (1 μg/mL), fucoidan (0.5 μg/mL) and Fuco-MnO_2_-NPs (1 μg/mL) for 3 h in hypoxic (1% oxygen) or normoxic conditions and then exposed to IR. Cell cycle analysis was performed using flow cytometry, following the staining of cells with PI after 72 h of incubation under hypoxic or normoxic conditions. The cells were collected by trypsinization, fixed in 70% ethanol at −20 °C, washed in PBS, resuspended in 1 mL of PBS containing 1 mg/mL of ribonuclease and 50 µg/mL of PI, incubated in the dark for 30 min at 37 °C, and analyzed by flow cytometry (FACSVerse™, Becton-Dickinson, San Jose, CA, USA).

### 4.9. Aapoptosis Assay

The cells (1 × 10^5^) were plated in six-well plates and allowed to attach overnight. The cells were exposed to hypoxic (1% oxygen) and normoxic conditions with PAH-MnO_2_-NPs (1 μg/mL), fucoidan (0.5 μg/mL), and Fuco-MnO_2_-NPs (1 μg/mL) for 3 h and then exposed to IR. After 72 h of irradiation, the extent of apoptosis was evaluated by Annexin V-FITC and flow cytometry. Briefly, the cells were treated with trypsin, washed with PBS (pH 7.4), and stained with Annexin V-FITC (BD Biosciences, San Diego, CA, USA) and 2 μg/ml of PI in 100 μL of Annexin V binding buffer (10 mM HEPES, pH 7.4/140 mM NaCl/2.5 mM CaCl_2_) for 15 min at 37 °C in the dark. The samples were analyzed by flow cytometry using a BD FACSVerse™ flow cytometer (BD Biosciences, San Jose, CA, USA). Data analysis was performed using the BD FACSuite™ software (BD Biosciences, San Jose, CA, USA).

### 4.10. DNA Damage Repair Analysis

DNA damage kinetics were analyzed using two approaches: flow cytometry and cell imaging of γ-H2AX immunofluorescence. For flow cytometry, 2 × 10^5^ cells were plated in six-well plates and allowed to attach overnight. The cells were treated with PAH-MnO_2_-NPs (1 μg/mL), fucoidan (0.5 μg/mL), and Fuco-MnO_2_-NPs (1 μg/mL) for 3 h and subsequently exposed to IR under normoxia or hypoxia conditions. At 2 h or 24 h after irradiation, cells were collected by trypsinization and fixed in 4% formaldehyde for 10 min, followed by permeabilization with 0.01% Triton X-100 for three min. Blocking was performed in 2% FBS in PBS for 30 min at room temperature, followed by incubation for one hour at 25 °C with γ-H2AX antibody. Secondary antibodies were added and incubated for 30 min. The samples were analyzed by flow cytometry, and data analysis was performed with BD FACSVerse™ (BD Biosciences) and BD FACSuite™ software (BD Biosciences). Negative control staining was performed with secondary antibodies alone.

For cell imaging, cells were cultured on a cover glass (Paul Marienfeld GmbH & Co. KG, Lauda-Königshofen, Germany) and then fixed with 4% formaldehyde and permeabilized using 0.01% Triton X-100. After being blocked with 2% FBS for 30 min, cells were incubated with γ-H2AX antibody (Millipore and Cell Signaling Technology, Danvers, MA, USA) for one hour, followed by Alexa-Fluor-488-conjugated secondary antibody (Life Technologies Corporation, Carlsbad, CA, USA) and DAPI (Sigma-Aldrich) application for 30 min. The cells were then washed, mounted using glycerol, and analyzed by fluorescent microscopy (Zeiss Observer D1; Carl Zeiss Co., Ltd.). The nuclei were segmented and counted based on DAPI staining, while γ-H2AX foci were counted and expressed as average foci per nucleus. The quantification of each independent experiment included approximately 45 cells.

### 4.11. Animal Model

All animal procedures were conducted in accordance with all appropriate regulatory standards under protocol (ID: 20160909001; approval date: 2016-09-26) approved by the Institutional Animal Care and Use Committee of Samsung Biomedical Research Institute. Six- to seven-week-old male Balb/c mice were purchased from Orient Bio (Gapyeong, Republic of Korea). BxPC-3 cells (1 × 10^6^ cells/50 μL PBS) were injected subcutaneously into the right hind leg. Tumor volumes were measured every three days with calipers and calculated according to the following formula: volume = DShort^2^ × DLong ÷ 2. When the mean tumor volume reached 80–150 mm^3^, the mice were randomized. Nanoparticles (200 ng/50 µL) and fucoidan (100 ng/50 µL) were administered by intratumoral injection weekly, beginning on the day of randomization. The injections were continued until the day prior to sacrifice. 4 h after drug treatment, the tumor was irradiated with 8 Gy of X-ray to the right hind leg. During irradiation, the mice were anesthetized by intraperitoneal injection of 30 mg/kg of Zoletil (Virbac, Carros, France) and 10 mg/kg of Rompun (Bayer, Leverkusen, Germany) under prescription of a veterinarian. The animals were housed under barrier conditions and fed a standard rodent diet and water. At the end of the experiment, tumor tissue was fixed with 10% neutral buffered formalin (NBF; Sigma-Aldrich, St. Louis, MO, USA) and embedded in paraffin for IHC staining.

### 4.12. TUNEL Assay

Apoptosis in the tumor tissues was determined using the TUNEL assay. The tumor tissues were fixed with 10% NBF for 4 h and then embedded in paraffin. The slices were deparaffinized in water and placed in 3% H_2_O_2_ for 10 min at room temperature. TUNEL staining was done using the ApopTag^®^ Peroxidase In Situ Apoptosis Detection Kit (Millipore, Burlington, MA, USA). Images were captured using a digital pathology slide scanner (Aperio ScanScope AT; Leica Biosystems, Buffalo Grove, IL, USA). The numbers of TUNEL-positive cells were determined with the ImageScope software (Leica Biosystems, Buffalo Grove, IL, USA).

### 4.13. Immunohistochemical Analysis

Immunohistochemical studies were carried out on formalin-fixed and paraffin-embedded, 4 µm-thick tissue sections. Tissue sections were deparaffinized three times in xylene for a total of 15 min and subsequently rehydrated. Immunostaining was performed using a Bond-maxTM Polymer refines detection kit (Vision Biosystems, Melbourne, Australia). Briefly, antigen retrieval was carried out at 97 °C for 20 min in an ER1 buffer. After blocking the endogenous peroxidase activity with 3% hydrogen peroxidase for 10 min, incubation with primary antibodies was carried out for 15 min at room temperature at a dilution of 1:200. The primary antibodies were anti-HIF-1α (Novus Biologicals, Littleton, CO, USA) and anti-phospho-VEGF receptor 2, and anti-CD31 (Abcam) was carried out for 15 min at room temperature at a dilution of 1:200.

### 4.14. Statistical Analysis

The significance of differences between the two groups was calculated with Student’s t-tests in Microsoft Excel (Microsoft Corp., Redmond, WA, USA). The significance of differences between more than two groups was calculated with two-way analysis of variance tests in GraphPad Prism 7 (GraphPad Software, La Jolla, CA, USA). *p* values < 0.05 were considered statistically significant.

## 5. Conclusions

Hypoxia is a well-recognized mechanism of radioresistance, leading to tumor recurrence after RT. In this study, we demonstrated that MnO_2_-NPs efficiently sensitized pancreatic cancer cells to RT both in vitro and in vivo by alleviating hypoxia. Fucoidans, which were conjugated to NPs, exerted antiangiogenic effect in vivo. Our findings suggest a potential of the clinical use of this new nanoplatform in the treatment of hypoxic, radioresistant pancreatic cancer.

## Figures and Tables

**Figure 1 marinedrugs-16-00510-f001:**
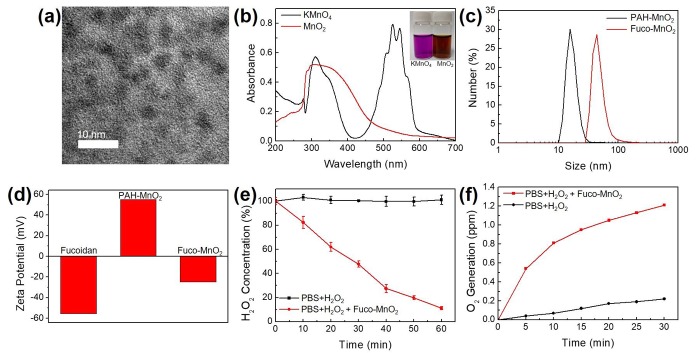
Synthesis of fucoidan-coated manganese dioxide nanoparticles (Fuco-MnO_2_-NPs) and their reactivity toward endogenous hydrogen peroxide (H_2_O_2_). (**a**) Transmission electron microscopy (TEM) image of MnO_2_-NPs. (**b**) Ultraviolet-visible (UV-Vis) absorption spectra of KMnO_4_ solution and MnO_2_-NPs. A broad peak at approximately 300 nm indicated the formation of MnO_2_-NPs. (**c**) Hydrodynamic size distribution of MnO_2_-NPs before and after fucoidan coating. (**d**) Effect of fucoidan coating on zeta-potential of MnO_2_-NPs. (**e**) Decomposition of H_2_O_2_ by Fuco-MnO_2_-NPs (*n* = 3). (**f**) Generation of oxygen after treatment with Fuco-MnO_2_-NPs.

**Figure 2 marinedrugs-16-00510-f002:**
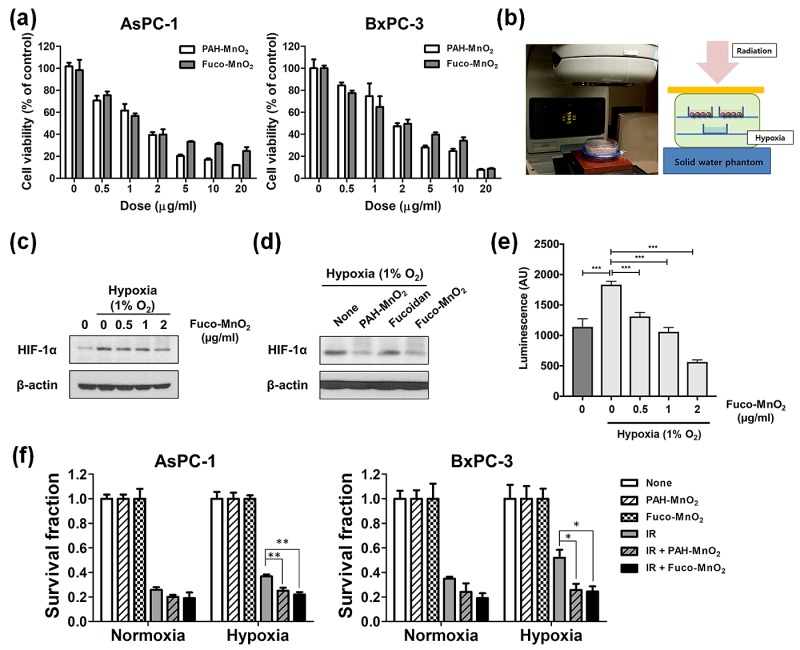
In vitro radiosensitizing effect of fucoidan-coated manganese dioxide nanoparticles (Fuco-MnO_2_-NPs) in human pancreatic cancer cells. (**a**) AsPC-1 and BxPC-3 cells were cultured in complete medium and treated with different concentrations of PAH-MnO_2_-NPs or Fuco-MnO_2_-NPs (0, 0.5, 1, 2, 5, 10 and 20 μg/mL) for 48 h. Cell viability was assessed using the cell counting kit 8 (CCK-8 kit). Data are presented as mean ± standard deviation (SD) normalized to the values for an untreated control (*n* = 4). (**b**) Schematic view of irradiation setup. Simulated hypoxia was established by culturing human pancreatic cancer cells in a modular incubator chamber flushed with 1% oxygen. (**c**) Fuco-MnO_2_-NPs inhibited hypoxia-driven HIF-1α expression in a dose-dependent manner. (**d**) PAH-MnO_2_-NPs and Fuco-MnO_2_-NPs but not fucoidan inhibited HIF-1α expression under hypoxia. (**e**) Fuco-MnO_2_-NPs inhibited luciferase reporter activity driven by hypoxia-response elements (*n* = 3). (**f**) Radiosensitivity was determined by clonogenic survival assay. AsPC-1 and BxPC-3 cells were pretreated with 10 μg/mL of either PAH-MnO_2_-NPs or Fuco-MnO_2_-NPs, followed by 4 Gy of X-ray irradiation. Colonies with >50 cells were counted. Data are presented as mean ± SD (*n* = 4); * *p* < 0.05; ** *p* < 0.01.

**Figure 3 marinedrugs-16-00510-f003:**
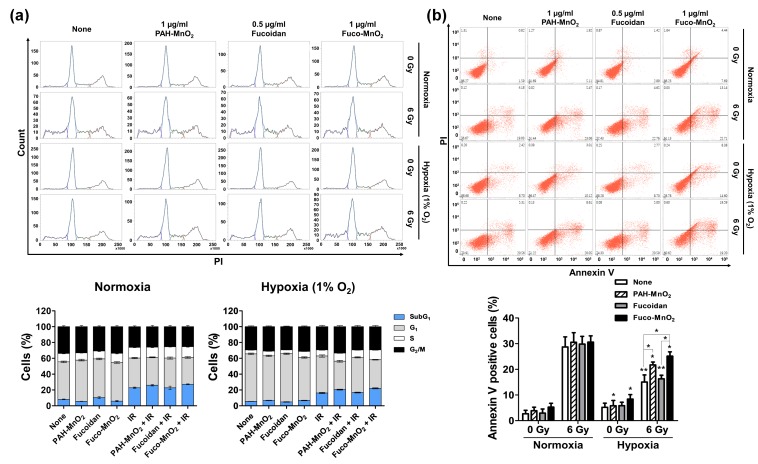
Effects of fucoidan-coated manganese dioxide nanoparticles (Fuco-MnO_2_-NPs) on ionizing radiation (IR) -induced apoptotic cell death in BxPC-3 cells. (**a**) Flow cytometry evaluation of cell cycle progression at 72 h post-irradiation. BxPC-3 cells were pretreated with 0.5 μg/mL of fucoidan and 1 μg/mL of PAH-MnO_2_-NPs and Fuco-MnO_2_-NPs for 3 h in a hypoxic (1% oxygen) or normoxic condition and then exposed to IR. 6 Gy of X-ray was applied because 4 Gy was not strong enough to induce apoptosis in BxPC-3 cells. After 72 h, cells were fixed and stained with PI. Histograms show the representative DNA content stained with PI. Stacked bar graphs show the relative percentage of cells at different cell cycle phases (*n* = 3). (**b**) Apoptosis was assessed by percentage of Annexin-V-positive cells using flow cytometry. PAH-MnO_2_-NPs and Fuco-MnO_2_-NPs reversed suppression of IR-induced apoptosis by hypoxia (*n* = 3). Data are presented as mean ± SD; * *p* < 0.05; ** *p* < 0.01.

**Figure 4 marinedrugs-16-00510-f004:**
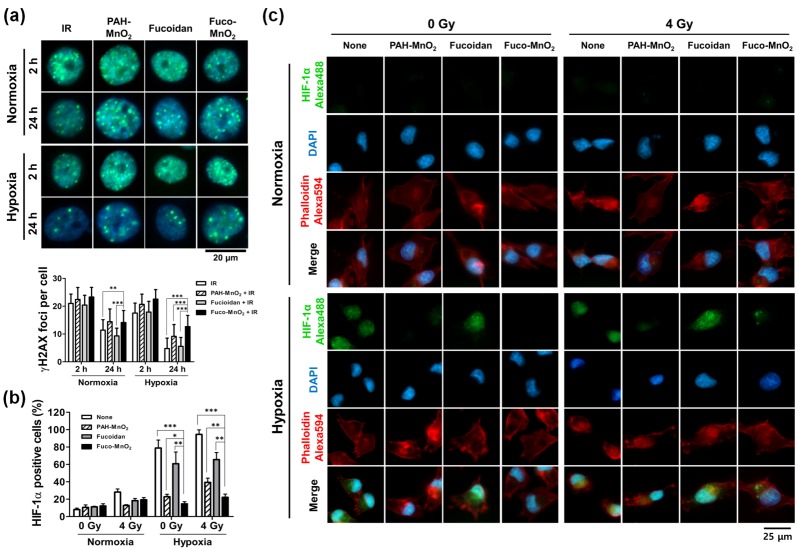
Effects of fucoidan-coated manganese dioxide nanoparticles Fuco-MnO_2_-NPs on DNA damage repair and hypoxia-inducible factor 1 (HIF-1α) expression after exposure to ionizing radiation IR. (**a**) Representative immunofluorescence images of γ-H2AX foci in BxPC-3 cells at 2 h and 24 h postirradiation (upper panel). BxPC-3 cells were pretreated with fucoidan, PAH-MnO_2_-NPs, and Fuco-MnO_2_-NPs for 3 h, followed by 4 Gy of IR under hypoxic or normoxic conditions. The γ-H2AX foci were seen in green, and the nuclei were counterstained with DAPI. Quantification data (lower panel) show that MnO_2_-NPs attenuated DNA damage repair; *n* = 40. (**b**) Flow cytometric analysis of HIF-1α-positive BxPC-3 cells. The expression of HIF-1α was induced by hypoxia and was significantly decreased by PAH-MnO_2_-NPs and Fuco-MnO_2_-NPs. Data are presented as mean ± SD (*n* = 3); * *p* < 0.05; ** *p* < 0.01. (**c**) Nuclear expression of HIF-1α detected by immunofluorescence at 24 h post-irradiation. Representative images are shown. DAPI in blue; F-actin in red; HIF-1α in green.

**Figure 5 marinedrugs-16-00510-f005:**
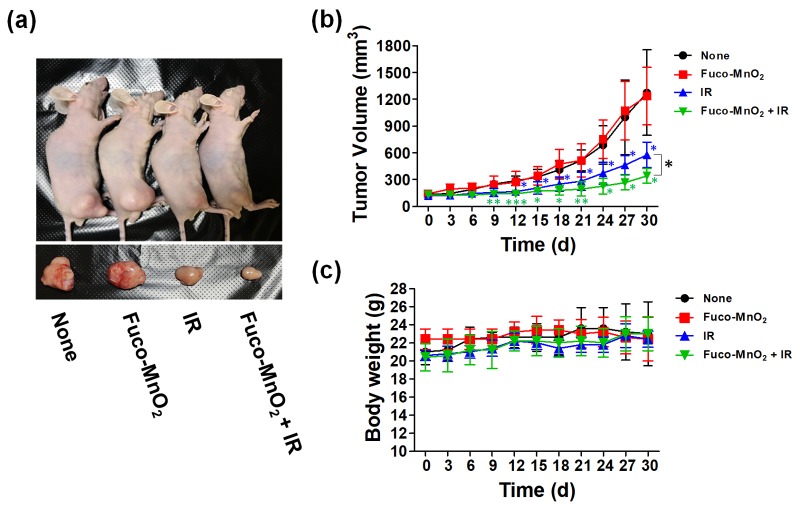
In vivo radiosensitizing efficacy of fucoidan-coated manganese dioxide nanoparticles (Fuco-MnO_2_-NPs). (**a**) Pictures of tumors collected from BxPC-3 tumor-bearing BALB/c nude mice. On Day 30, the mice were sacrificed. (**b**) Fuco-MnO_2_-NPs did not affect BxPC-3 tumor growth. Ionizing radiation (IR) significantly reduced tumor growth, which was further inhibited by co-treatment with Fuco-MnO_2_-NPs. The BxPC-3 cells were implanted into the right legs of BALB/c nude mice. When tumors were palpable (80–150 mm^3^), three groups of mice (*n* = 5 in a group) received either Fuco-MnO_2_-NPs (200 ng/50 μL/week, intratumoral injection), 8 Gy of IR, or the combination of Fuco-MnO_2_-NPs and IR. Tumor volume was calculated as detailed in the Materials and Methods section. Data are presented as mean ± SD; * *p* < 0.05; ** *p* < 0.01; *** *p* < 0.001. (**c**) Changes in body weight of the four groups were monitored.

**Figure 6 marinedrugs-16-00510-f006:**
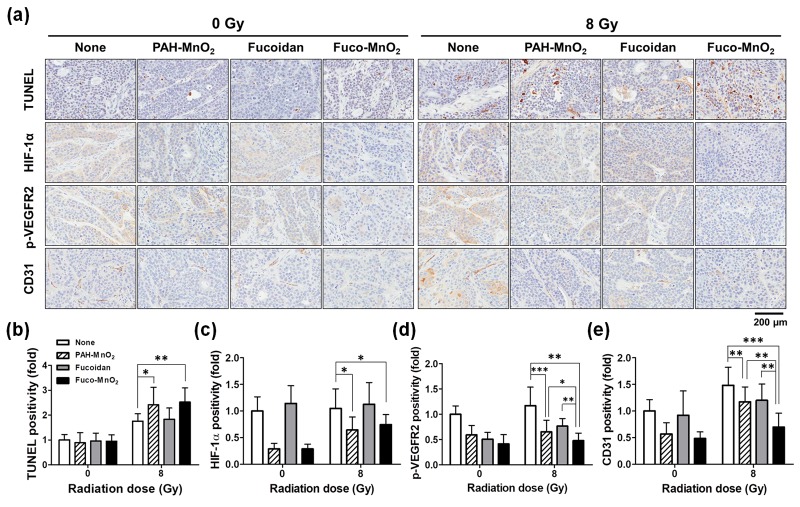
Effects of fucoidan-coated manganese dioxide nanoparticles (Fuco-MnO_2_-NPs) on terminal deoxynucleotidyl transferase dUTP nick end-labeling (TUNEL), hypoxia-inducible-factor-1 (HIF-1α), vascular endothelial growth factor receptor 2 (phospho-VEGFR2), and cluster of differentiation 31 (CD31) expression in BxPC-3 xenograft tumor tissues. (**a**) Representative images of TUNEL and IHC. Tumors were harvested on Day 30 after ionizing radiation (IR) treatment. Formalin-fixed and, paraffin-embedded BxPC3 tumor tissues were further analyzed for immunohistochemistry (IHC). (**b**) Quantification data show that IR-induced apoptosis was further enhanced by Fuco-MnO_2_-NPs, as assessed by TUNEL assays (*n* = 10), and that Fuco-MnO_2_-NPs inhibited the expression of HIF-1α (*n* = 9) (**c**), phospho-VEGFR2 (*n* = 9) (**d**), and CD31 (*n* = 9) (**e**). For all graphs, data are presented as mean ± SD; * *p* < 0.05; ** *p* < 0.01; *** *p* < 0.001.

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
