# Peer review of "Fucoidan-Manganese Dioxide Nanoparticles Potentiate Radiation Therapy by Co-Targeting Tumor Hypoxia and Angiogenesis"

_marinedrugs, 2018, doi:10.3390/md16120510_

Round 1

Reviewer 1 Report

In this manuscript Shin and colleagues developed a new nanoparticle (NP) platform (MnO2 NPs coated with fucoidan) in order to address the deficiency associated with radiation therapy effectiveness in prostate cancer. The authors hypothesize that this deficiency can mainly be addressed by increasing oxygen availability within the tumor using this NP platform. While the data is generally supportive of the authors conclusions, some concerns are listed below:

1. There are multiple figures in which I would like to see more controls done. Some figures leave out the fucoidan (the drug) by itself control, while others leave out the PAH control NP. For example, in Figure 2c I would like to see how the PAH control NP affects HIF1α protein expression in the cell line. This is mainly because in Figure 6 there is no difference in HIF1α staining in the mouse between PAH and Fuco NPs. Furthermore, since MnO2 NPs have been shown to produce oxygen, this affect may be due to the NP and not the fucoidan drug.

2. Some in vitro experiments were done in both AsPC-1 and BxPC-3 and others were only done with a single cell line (BxPC-3).

3. In some figures the control is not defined. For example, in Figure 5 is the control no treatment, the PAH-MnO2 control NP, or fucoidan itself?

4. The amount of radiation for in vitro experiments changes between experiments and is not explained why it changes even though all other experimental conditions are the same.

5. In Figure 4A, under normoxic conditions the PAH control and Fuco-NPs show almost identical numbers and from what I can see on the images have almost identical H2AX foci. However, these bars are significantly different. Please address this.

a. Furthermore, are all of these data shown with standard error (SE) and not standard deviation (SD)?

6. In a few places the authors seem to overstate their data. This is highlighted in some cases where the PAH control and Fuco-NPs display similar results and the authors claiming that their Fuco NPs have an effect greater than the control.

Also, some minor points to address:

1. I could not find mention of Figure 1F.

2. In Figure 5, the text (results) and figure legend mention 6Gy, however the text (methods) and the figure legend in Figure 5B and 5C mention 8Gy treatment. This needs to be addressed.

3. Image quality for Figure 4A and 4C are hard to see and are therefore hard to determine what the authors are claiming.

4. The number of samples is missing from many of the figures: Figure 2, Figure 3, Figure 4

5. The discussion mainly restates the results and needs to address more the uses that this platform provides for research/clinic/etc.

Author Response

Reviewer #1:

Summary Comments: “In this manuscript Shin and colleagues developed a new nanoparticle (NP) platform (MnO2 NPs coated with fucoidan) in order to address the deficiency associated with radiation therapy effectiveness in prostate cancer. The authors hypothesize that this deficiency can mainly be addressed by increasing oxygen availability within the tumor using this NP platform. While the data is generally supportive of the authors conclusions, some concerns are listed below:”

Our response:

We truly appreciate the reviewer for reviewer’s consideration on our work and insightful comments. In accordance with his or her comments, we have revised the manuscript in point-by-point manner.

Comment #1: “There are multiple figures in which I would like to see more controls done. Some figures leave out the fucoidan (the drug) by itself control, while others leave out the PAH control NP. For example, in Figure 2c I would like to see how the PAH control NP affects HIF1α protein expression in the cell line. This is mainly because in Figure 6 there is no difference in HIF1α staining in the mouse between PAH and Fuco NPs. Furthermore, since MnO2 NPs have been shown to produce oxygen, this affect may be due to the NP and not the fucoidan drug.”

Our response: As requested, we included a new figure (Fig.1d) showing that both PAH-control and Fuco-NPs inhibited HIF1α protein expression under hypoxic condition. This is well consistent with HIF1α staining data in the mouse tissue samples, as shown in Fig. 6a and c. As explained by the reviewer, the in vitro and in vivo HIF1α staining data indicate that hypoxia is relieved by the NPs, but not by the fucoidan drug. Fucoidan drug did not affect HIF1α protein expression in both cultured cells under hypoxia (Fig. 1d) and tumor tissues (Fig. 6c), but rather decreased angiogenesis on the tumor tissue samples.

Comment #2: “Some in vitro experiments were done in both AsPC-1 and BxPC-3 and others were only done with a single cell line (BxPC-3).”

Our response: We initially tested our hypothesis in three pancreatic cancer cell lines including MIA PaCa-2 cell line and obtained the similar findings. Because BxPC-3 was the best in vivo model, we focused on BxPC-3. We explained this in the revised text.

Comment #3: “In some figures the control is not defined. For example, in Figure 5 is the control no treatment, the PAH-MnO2 control NP, or fucoidan itself?”

Our response: We agree with the reviewer’s comment. As requested, we carefully revised the figure legends to clearly define what experimental control was used.

Comment #4: “The amount of radiation for in vitro experiments changes between experiments and is not explained why it changes even though all other experimental conditions are the same.”

Our response: We used 4 Gy of X-rays as a standard in vitro treatment, but the 4 Gy was not strong enough to induce apoptotic death under normoxia. We were able to see the hypoxia-mediated suppression of apoptosis when 6 Gy was applied. We explain this in the revised text.

Comment #5: “ In Figure 4A, under normoxic conditions the PAH control and Fuco-NPs show almost identical numbers and from what I can see on the images have almost identical H2AX foci. However, these bars are significantly different. Please address this.

a. Furthermore, are all of these data shown with standard error (SE) and not standard deviation (SD)?”

Our response: We agree with the reviewer’s comments. We re-analyzed data and found that there was no significant difference between PAH control and Fuco-NPs (p = 0.08), which is now corrected in the revised figure. We now present all the data with standard deviation.

Comment #6: “In a few places the authors seem to overstate their data. This is highlighted in some cases where the PAH control and Fuco-NPs display similar results and the authors claiming that their Fuco NPs have an effect greater than the control.”

Our response: We agree with the reviewer’s comment that there is no difference between PAH control and Fuco-NPs in in vitro data. Fuco-NPs have a greater effect than the PAH control only regarding anti-angiogenic activity. We revised the text accordingly.

Minor Comment #1: “I could not find mention of Figure 1F.”

Our response: We agree with the reviewer’s comment and mentioned Figure 1f in the revised text.

Minor Comment #2: “In Figure 5, the text (results) and figure legend mention 6Gy, however the text (methods) and the figure legend in Figure 5B and 5C mention 8Gy treatment. This needs to be addressed.”

Our response: We used a single dose of 8 Gy for animal experiment. We corrected them in the revised text.

Minor Comment #3: “Image quality for Figure 4A and 4C are hard to see and are therefore hard to determine what the authors are claiming.”

Our response: As requested, we replaced Figure 4a and 4c with better quality images. For Figure 4c, we now present separate color images for HIF-1, F-actin and DAPI.

Minor Comment #4: “The number of samples is missing from many of the figures: Figure 2, Figure 3, Figure 4g.”

Our response: As requested, we included the numbers of samples in Fig 2, Fig3 and Fig4.

Minor Comment #5: “The discussion mainly restates the results and needs to address more the uses that this platform provides for research/clinic/etc..”

Our response: As requested, we revised the discussion to address the use of our platform for research and clinic.

Thank you very much for the valuable comments. Your comments significantly improved the quality of the manuscript.

Reviewer 2 Report

The manuscript by Sung-Won Shin et al. reports on the evaluation of the possibility to potentiate Radiation Therapy by using Fucoidan–Manganese Dioxide Nanoparticles.

The manuscript is well organized and presented, and the results are in agreement with author hypothesis. Thus, the evaluation of this reviewer is that the manuscript is suitable for publication in Marine Drugs. I would only suggest minor revision to further improve the paper. In details, abstract should be emended by inserting some key results and in the discussion sections authors should compare their results with some available literature data to strengthen the advantages and the importance of their approach in the field.

Author Response

Reviewer #2:

Summary Comments: “The manuscript by Sung-Won Shin et al. reports on the evaluation of the possibility to potentiate Radiation Therapy by using Fucoidan–Manganese Dioxide Nanoparticles.

The manuscript is well organized and presented, and the results are in agreement with author hypothesis. Thus, the evaluation of this reviewer is that the manuscript is suitable for publication in Marine Drugs. I would only suggest minor revision to further improve the paper. In details, abstract should be emended by inserting some key results and in the discussion sections authors should compare their results with some available literature data to strengthen the advantages and the importance of their approach in the field.

Our response:

We truly appreciate the reviewer for reviewer’s consideration on our work and insightful comments. As the reviewer suggested, we added discussion/comparison with the analogous type of NPs with similar features.

Our modification to the manuscript.

Hyperbaric oxygen therapy is a well-established treatment to overcome hypoxia, where pure O2 is supplied in a pressurized room. However, this may cause overproduction of ROS in normal tissue, resulting in serious side effects such as barotrauma and hyperoxic seizures [25]. In addition, O2 molecule can be directly delivered using its high solubility for perfluorocarbon [26]. Unfortunately, the size of perfluorocarbon-based emulsion is too large, which limits penetration into a hypoxic region. One of the strategies to deliver oxygen into hypoxic tumor tissues is to use oxygen-generating nanomaterials, such as CaO2 NPs [27], C3N4 NPs [28], and catalase-containing emulsion [29]. Recent studies have reported that MnO2-NPs can convert H2O2, highly produced in tumor tissues, into oxygen, thus modulating tumor hypoxia and enhancing radiation efficacy in mouse models [13-15]. In these studies, MnO2-NPs are coated with biologically inactive albumin.

Thank you very much for the valuable comments. Your comments significantly improved the quality of the manuscript.

Reviewer 3 Report

In the manuscript entitled "Fucoidan–Manganese Dioxide Nanoparticles Potentiate Radiation Therapy by Cotargeting Tumor Hypoxia and Angiogenesis" the authors report an interesting investigation of Fucoidan-loaded Manganese dioxide nanoparticles enhancement with irradiation to co-target Hypoxia and Angiogenesis. Nanoparticles of Manganese Dioxide were developed and conjugated with Fucoidan as antitumour and antiangiogenic agent. By in vitro and in vivo study Shin and co-workers could demonstrate that the combination with radiation therapy and Fuco-MnO2 NPs leads to a synergistic targeting of tumour hypoxia and angiogenesis.These results deserve publication after (major) revision. Several points needs to be developed \clarified, The crtical points are listed below:

1) A moderate English revision by mother tongue is suggested

2) A list of acronyms (such as a table) shoudl be added for better readability of paper. Please add it.

3) Results and discussion suffer from some weak points not developed\discussed such as a comprehensive comparison\discussion with analogous type of NPs (eg inhorganic or organic ) having similar features. Please add\improve discussion.

4) Following point 3 an update of references is an obvious consequence, since they are surely overlooked

5) Co-localisation\uptake quantification: a FACS or CLSM analysis is conceivable: please add discussion\update these parts

6) Higher resolution SEM\TEM or AFM images for NPs characterisation are also suggested to be provided

7) Conclusions and outlook are missing, please provide them

8) For the high ranked quality of journal a general improvement of figures resolution is recommended to reach standard for publication

Author Response

Reviewer #3:

Summary Comments: “In the manuscript entitled "Fucoidan–Manganese Dioxide Nanoparticles Potentiate Radiation Therapy by Cotargeting Tumor Hypoxia and Angiogenesis" the authors report an interesting investigation of Fucoidan-loaded Manganese dioxide nanoparticles enhancement with irradiation to co-target Hypoxia and Angiogenesis. Nanoparticles of Manganese Dioxide were developed and conjugated with Fucoidan as antitumour and antiangiogenic agent. By in vitro and in vivo study Shin and co-workers could demonstrate that the combination with radiation therapy and Fuco-MnO2 NPs leads to a synergistic targeting of tumour hypoxia and angiogenesis. These results deserve publication after (major) revision. Several points needs to be developed \clarified, The crtical points are listed below:”

Our response:

We truly appreciate the reviewer for reviewer’s consideration on our work and insightful comments. In accordance with his or her comments, we have revised the manuscript in point-by-point manner.

Comment #1: “A moderate English revision by mother tongue is suggested”

Our response: As requested, our manuscript was carefully revised by English editing service.

Comment #2: “A list of acronyms (such as a table) shoudl be added for better readability of paper. Please add it.”

Our response: As requested, we included a list of all acronyms as a new table (Table 1) at the end of main text.

Comment #3: “Results and discussion suffer from some weak points not developed\discussed such as a comprehensive comparison\discussion with analogous type of NPs (eg inhorganic or organic ) having similar features. Please add\improve discussion.”

Our response: We appreciate reviewer’s insightful comment. As the reviewer suggested, we added discussion on the analogous types of NPs.

Our modification to the manuscript:

Hyperbaric oxygen therapy is a well-established treatment to overcome hypoxia, where pure O2 is supplied in a pressurized room. However, this may cause overproduction of ROS in normal tissue, resulting in serious side effects such as barotrauma and hyperoxic seizures [25]. In addition, O2 molecule can be directly delivered using its high solubility for perfluorocarbon [26]. Unfortunately, the size of perfluorocarbon-based emulsion is too large, which limits penetration into a hypoxic region. One of the strategies to deliver oxygen into hypoxic tumor tissues is to use oxygen-generating nanomaterials, such as CaO2 NPs [27], C3N4 NPs [28], and catalase-containing emulsion [29]. Recent studies have reported that MnO2-NPs can convert H2O2, highly produced in tumor tissues, into oxygen, thus modulating tumor hypoxia and enhancing radiation efficacy in mouse models [13-15]. In these studies, MnO2-NPs are coated with biologically inactive albumin.

Comment #4: “Following point 3 an update of references is an obvious consequence, since they are surely overlooked.”

Our response: We thank the reviewers for valuable comments. We added the reference the reviewer suggested.

Our modification to the manuscript:

[25] Chen, Q.; Huang, Z.; Chen, H.; Shapiro, H.; Beckers, J.; Hetzel, F.W. Improvement of tumor response by manipulation of tumor oxygenation during photodynamic therapy. Photochem. Photobiol. 2002, 76, 197−203.

[26] Song, X.; Feng, L.; Liang, C.; Yang, K.; Liu, Z. Ultrasound Triggered Tumor Oxygenation with Oxygen-Shuttle Nanoperfluorocarbon to Overcome Hypoxia-Associated Resistance in Cancer Therapies. Nano Lett. 2016, 16, 6145−6153.

[27] Huang, C.C.; Chia, W.T.; Chung, M.F.; Lin, K.J.; Hsiao, C.W.; Jin, C.; Lim, W.H.; Chen, C.C.; Sung, H.W. An Implantable Depot That Can Generate Oxygen in Situ for Overcoming Hypoxia-Induced Resistance to Anticancer Drugs in Chemotherapy. J. Am. Chem. Soc. 2016, 138, 5222−5225.

[28] Zheng, D.W.; Li, B.; Li, C.X.; Fan, J.X.; Lei, Q.; Li, C.; Xu, Z.; Zhang, X. Z. Carbon-Dot-Decorated Carbon Nitride Nanoparticles for Enhanced Photodynamic Therapy against Hypoxic Tumor via Water Splitting. ACS Nano 2016, 10, 8715−8722.

[29] Chen, H.; Tian, J.; He, W.; Guo, Z. H2O2Activatable and O2Evolving Nanoparticles for Highly Efficient and Selective Photodynamic Therapy against Hypoxic Tumor Cells. J. Am. Chem. Soc. 2015, 137, 1539−1547.

Comment #5: Co-localisation\uptake quantification: a FACS or CLSM analysis is conceivable: please add discussion\update these parts.”

Our response: We thank the reviewers for valuable comments. But, we think that MnO2-NPs outside the cells also contribute to relive hypoxia since O2 molecules can diffuse across the cellular membranes. Therefore, quantification of cellular uptake may overestimate the role of MnO2-NPs.

Comment #6: “Higher resolution SEM\TEM or AFM images for NPs characterisation are also suggested to be provided.”

Our response: We thank the reviewer for valuable comments. As the reviewer suggested, an HR-TEM image of nanoparticles was added to show the structure of nanoparticles clearly (Fig. 1a).

Comment #7: “Conclusions and outlook are missing, please provide them.”

Our response: As requested, we included a conclusion section in the revised manuscript.

Comment #8: “For the high ranked quality of journal a general improvement of figures resolution is recommended to reach standard for publication.”

Our response: As requested, we improved the quality and resolution of figure images in the revised manuscript.

Thank you very much for the valuable comments. Your comments significantly improved the quality of the manuscript.

Round 2

Reviewer 1 Report

My comments have been addressed. The paper reads much better and is suitable for publication now.

Reviewer 3 Report

The authors have successfully addressed all issues raised by previous report in a way that the manuscript can be accepted in present form.